# Different Liquid Biopsies for the Management of Non-Small Cell Lung Cancer in the Mutational Oncology Era

**DOI:** 10.3390/medsci11010008

**Published:** 2023-01-04

**Authors:** Maria Palmieri, Elisa Frullanti

**Affiliations:** Med Biotech Hub and Competence Center, Department of Medical Biotechnologies, University of Siena, 53100 Siena, Italy

**Keywords:** lung cancer, liquid biopsy, non-small cell lung carcinoma

## Abstract

In the last ten years, liquid biopsy has been slowly joining the traditional invasive techniques for the diagnosis and monitoring of tumors. Liquid biopsies allow easy repeated sampling of blood, reflect the tumor scenario, and make personalized therapy real for the patient. Liquid biopsies isolate and utilize different substrates present in patients’ body fluids such as circulating tumor cells, circulating tumor DNA, tumor extracellular vesicles, etc. One of the most-used solid cancers in the development of the non-invasive liquid biopsy approach that has benefited from scientific advances is non-small cell lung cancer (NSCLC). Using liquid biopsy, it is possible to have more details on NSCLC staging, progression, heterogeneity, gene mutations and clonal evolution, etc., basing the treatment on precision medicine as well as on the screening of markers for therapeutic resistance. With this review, the authors propose a complete and current overview of all different liquid biopsies available to date, to understand how much has been carried out and how much remains to be completed for a better characterization of NSCLC.

## 1. Introduction

According to World Health Organization (WHO) data, globally, lung cancer accounts for 18% of annual cancer deaths in both sexes at all ages [1]. Non-small cell lung cancer (NSCLC) accounts for approximately 87% of all lung cancer patients [2]. Among these, 40% are adenocarcinoma, 25–30% are squamous carcinoma and 10–15% are large cell carcinomas [3].

In the era of precision medicine, NSCLC has become an eminent example of how therapeutic decision making is based on the identification of specific biomarkers, called oncogenic drivers, designed by the National Comprehensive Cancer Network (NCCN) guidelines, such as the epidermal growth factor (EGFR), rearrangements of anaplastic lymphoma kinase (ALK), ROS proto-oncogene1 (ROS1), serine/threonine-protein kinase B-Raf (BRAF), ERBB2, MET proto-oncogene, receptor tyrosine kinase (MET) exon 14 skipping, RET proto-oncogene (RET) and PDL-1 (programmed death ligand 1) [4,5] present in approximately 30% of NSCLC patients [3,6,7]. More recently, the G12C missense mutation of the Kirsten rat sarcoma viral oncogene homolog (KRAS) gene has been added to these and new therapeutic agents (i.e., adagrasib, sotorasib) have been developed [8]. These advances in cancer genetics are leading to the birth of mutational oncology.

Unfortunately, over the years, it has been observed that the number of diagnoses in advanced stages of the disease (stages III and IV) have increased [9], and even if low-dose spiral computed tomography (LDCT) is currently 4 times more sensitive [10,11] than X-ray imaging, it does not detect the small lesions of lung cancer at the early stage [12,13]. For this reason, to date, histopathological investigation still remains the diagnostic gold standard. Despite this, even with acceptable adverse events, tissue biopsies often need to be repeated, which means additional stress and pain for patients [14].

At the same time, international scientific entities such as the College of American Pathologists (CAP), the International Association for the Study of Lung Cancer (IASLC) and the Association for Molecular Pathology (AMP) recommend the use of liquid biopsy to accompany tissue biopsy when the tissue sampling turns out to be inadequate or infeasible [15]. A liquid biopsy is defined as a test performed on a blood sample to search for circulating substrates coming from cancer cells in a non-invasive manner. Therefore, different types of liquid biopsy can be used for molecular diagnostic investigation through the collection of different body fluids, the most used of which is the peripheral blood sample, with minimally invasive methods. The different substrates obtained from body fluids, such as circulating tumor cells (CTCs), circulating tumor DNA (ctDNA), exosomes and circulating miRNAs, are used for diagnosis, prognosis and disease monitoring following clinical treatment. Given the study of the main substrates, it is possible to capture tumor heterogeneity as well as evaluate the molecular changes longitudinally due to genomic instability and treatment resistance [16,17,18,19]. Thus, given the growing need to identify the molecular drivers that characterize the tumor clonal evolution over time, in order to increase the targeted treatment of NSCLC, this review aims to discuss the different substrates of liquid biopsy by evaluating their advantages and limits, diagnostic performance and clinical applications.

## 2. Methods

### Search Strategy, Inclusion Criteria and Data Collection

We systematically searched the PubMed, Medline, Embase and Web of Science databases up to 22 August 2022 for studies reporting the diagnostic use of CTCs, ctDNA, miRNA and exosomes in NSCLC patients, using different combinations of the keywords: “Non-small cell lung cancer AND circulating tumor cells” or “Non-small cell lung cancer AND circulating tumor DNA”, “Non-small cell lung cancer AND miRNA liquid biopsy” or “Non-small cell lung cancer AND exosomes liquid biopsy”, filtering by publications in the last 10 years and with full-text availability. 

A total of 1827 potential studies were initially evaluated for the review on the results of the bibliographic search. Eligible publications were evaluated by checking the titles and abstracts and 120 full-text articles were selected for further evaluation of full text. Only studies published in English with full text were included. The inclusion criteria were all NSCLC patients studied with a liquid biopsy approach. In the process of assessing the eligibility, we firstly excluded case reports, method papers, comment/contribution papers, missing abstract and articles written in any language other than English. We subsequently excluded studies that involved cell lines or artificial samples. 

Finally, eligible studies were excluded from the final filters if they presented data in other tumor type or in a mixed way from different types of cancers other than the NSCLC, identifying 104 studies.

All records were reviewed by two authors independently (MP and EF) and reached a consensus at each eligible study. A flowchart of the literature selection is shown in Figure 1. 

## 3. Results

### 3.1. Circulating Tumor Cells—CTCs

Undoubtedly, one of the most studied liquid biopsy substrates is the circulating tumor cell (CTC), reported for the first time in 1869 by the Australian doctor Thomas Ashworth. CTC detection is widely used to predict the prognosis of various lung cancers, including small cell lung cancer, squamous cell lung cancer, lung adenocarcinoma and large cell lung cancer [20]. CTCs are released from the primary tumor and/or metastases by dispersing into the bloodstream and represent the intermediate stage between the primary tumor and metastases as determined by the ability of tumor cells to invade other organs leading to the formation of a metastatic lesion [21]. 

Basically, there are two methods for isolating CTCs: (i) label-dependent systems/methods based on the detection of specific surface markers of CTCs and (ii) label-independent systems/methods based on physical or biological properties of CTCs [22]. The first is dependent on the label, and is characterized by the use of specific markers such as the adhesion molecule of epithelial cells (Ep-CAM), the human epidermal growth factor receptor 2 (HER2) [23], mucin 1 (MUC1), cytokeratins, etc. [24,25]. However, due to their ability to undergo epithelial–mesenchymal transition (EMT), CTCs have different phenotypes that cannot express these markers, disabling this isolation method. To overcome this obstacle, several label-dependent methods using negative depletion of CD45-positive leukocytes have been introduced [26].

In contrast, label-independent methods isolate cells based on the size, density, specific electrical properties and invasive capabilities of the CTCs [22,24,26,27]. The advantage of these methods lies in their ability to isolate larger portions of CTCs because they are not bound by the expressed markers of the CTCs.

Once isolated, in NSCLC, the CTCs can be studied on the basis of their presence, quantity and their morphology, as well as by immunocytochemistry, genomic analysis, transcriptomics and proteomics [28].

Several studies have shown that the presence of CTCs is independent of the stage of the tumor and its histology [29,30]. Therefore, we deduce that the evaluation of the only number of CTCs in patients affected by NSCLC is incomplete information for clinical use.

The amount of CTCs in the blood is not high: it is estimated from 1 to 10 cells per 10 mL of blood due to the process of EMT [31] closely related to the higher stage and poorer prognosis of small cell lung cancer [32]. For this reason, there are preliminary studies showing that CTC sampling gives greater detection success if it is performed at the pulmonary vein level instead of in the peripheral blood [33,34]. In order to have greater clinical usefulness information, the quantity of CTCs must also be considered for the evaluation of prognosis. It is demonstrated how the count of CTCs at baseline is used as a predictor of progression-free survival, PFS, and overall survival, OS, but it is also used for monitoring treatment and the progress of the disease towards clinical relapse or remission. Thus, the count of CTCs during therapy could serve as an indication to modify the chemotherapy drug [35,36,37].

Morphology and immunohistochemistry are other characteristics that can be studied for more complete information. Morphology is necessary to distinguish the CTCs from the rest of the surrounding cells by distinguishing the size and shape of the nucleus, the structure of the chromatin, the nucleoli and more. Immunocytochemistry, on the other hand, is studied to evaluate cancer antigens, e.g., the expression of PD-L1 in the CTCs of NSCLC patients as an indication for treatment using PD-1/PD-L1 immune checkpoint inhibitors [38].

With genomic analysis, mutations in the DNA of CTCs are studied, leading to a significant clinical value for the evaluation of the treatment [39]. However, it is also possible to proceed with the study of RNA, mainly mRNA, through transcriptomic analysis. In fact, gene expression is in agreement with the number of mRNA transcripts [40]. Again, the data of a high expression of genes associated with chemoresistance should lead to a re-evaluation of the incompatible treatment. Then, through mass spectrometry or Western blotting, it is possible to carry out the proteomic analysis of the CTCs [41,42].

The advantage in clinical practice in the use of CTCs lies in the relatively inexpensive and non-invasive tool for the study of cancer. Cancer cell counts allow for real-time assessment of disease progression, and an increase in the number of CTCs is interpreted as a relapse and a decrease as a sign of remission, even before these can be seen with obvious clinical signs [36,43,44]. Through gene profiling it is possible to evaluate the clones of resistance to therapy [36], thus being able to adopt the right treatments, and it is possible to screen and detect the CTCs in the blood in the early stages of the tumor [45]. Several studies have shown the malignancy of CTCs identified in patients with nodules or bronchopneumopathy-affected patients who developed carcinoma after years of the identification of CTCs [44,46,47,48]. Finally, the screening combined with the study of immunohistochemistry or gene profiling can give answers on the origin of the primary tumor [49].

### 3.2. Circulating Tumor DNA—ctDNA

Tumor-derived biomarkers used for liquid biopsy can arise from several biofluids including plasma, serum, urine, saliva or exhaled breath condensate (EBC), and pleural and cerebrospinal fluid. Among the various biomarkers that can be obtained, the most widely used is undoubtedly the cell-free tumor DNA (ctDNA), genotyped as a tumor marker 17 years after its discovery in cancer patients in 1977 [50]. The ctDNA is released by the neoplastic cells in the body fluids in variable amounts. In particular, the amount of ctDNA dispersion increases with the stage and metastatic sites [38,51,52].

The short half-life of ctDNA of ≃1 hr makes it very suitable for measuring tumor burden in real-time in response to therapy. The challenge remains the detection of ctDNA in plasma versus the rest of the much more abundant wild-type cfDNA, released by non-tumoral cells [38]. The ctDNA detection is further influenced by the clinical condition of the patient, the timing of sampling collection and the site of metastases. Therefore, the analysis of ctDNA is influenced by several factors such as (i) sampling in ethylenediaminetetraacetic acid (EDTA) tubes or specific cfDNA storage tubes such as Streck (cell-free DNA BCT) [53]. The collection in the first type of tubes implies rapid processing of the sampling, while the second one that preserves the quality of small fragments of DNA, allows the holding of the blood sample as it lasts for several days at room temperature without the need for on-site processing; (ii) storage and transport of the blood sample [15]; (iii) timing between blood sampling and plasma extraction; and (iv) the amount of blood drawn from the patient, which is commonly established as 20 mL but is still not standardized [54].

In clinical practice, to date, the analysis of ctDNA for the treatment of NSCLC is based on the amplification of single gene loci and, in very few cases, on whole genome sequencing [55,56]. To date, digital PCR is the approach that best guarantees sensitivity in early-stage lung cancer [57]. However, common PCR-based approaches are only applicable when the patient’s driver mutation is well known. Hence, NGS was expanded to the study of ctDNA using small gene panels designed on specific hot spots, which allowed for the broadening of gene mutation screening [58]. The limitation of this technique is that only a small number of genes are interrogated and it is not possible to detect the number of copy variations or structural variants if the breakpoint sequence has not previously been characterized. Conversely, deep sequencing of the whole exome [59] or genome [60,61] can provide a complete ctDNA profile, but low sensitivity and high cost have to be taken into account.

The need for predictive biomarkers in NSCLC has therefore pushed research in implementing liquid biopsy through the study of ctDNA for the treatment of NSCLC. A meta-analysis study in 1017 patients from 10 studies suggested that the reduction in early ctDNA was associated with improved progression-free survival (PFS), overall survival (OS) and objective response rate (ORR) in ICI-treated patients with advanced NSCLC [62]. However, not all patients are able to benefit from this type of treatment due to drug resistance [63], disease progression [64] or adverse immune-related events (irAE) [65]. Several studies have been carried out in recent years such as that by Verzè et al. in which liquid biopsies were performed on 22 patients with advanced NSCLC on ctDNA extracted from different body fluids. The results confirmed that the plasma liquid biopsy was able to detect the mutation in EGFR but without detecting any advantage in the combination of the different sources such as urine and EBC [66]. The development of cfDNA assays and their implementation in clinical practice is a valuable option in cases where the amount of tissue is inadequate for mutation testing or in patients who refuse or are unable to undergo tissue biopsy.

To date, the Food and Drug Administration (FDA) has approved Guardant360^®^ CDx (Guardant Health, Redwood City, CA, USA, 2020), the first liquid biopsy NGS ctDNA assay for therapeutic decisions on the use of osimertinib, a second-generation tyrosine kinase inhibitor, in NSCLC patients with EGFR mutations [67]. In particular, the mutations detected in EGFR are in exon 18 (G719X) substitutions, exon 19 deletions, exon 20 insertions and substitutions (T790M, S768I), and exon 21 substitutions (L858R, L861Q). Despite the progress achieved, there are still limitations given to the exclusive use of ctDNA, especially for the diagnosis of NSCLC in the initial phase, as few feasible quantities of plasma are required to guarantee a sufficient number of copies of ctDNA mutations [68].

### 3.3. microRNA-miRNA

miRNAs, also called microRNAs, are a class of small, single endogenous RNAs with a length of about 20–22 molecules that do not code for any protein. They act as post-transcriptional gene regulators by binding to the complementary 3′- untranslated regions (3′-UTR) of target mRNAs and causing translational inhibition or mRNA degradation [69]. The first discovery of miRNAs dates back to 1993 by Ambros and colleagues from Caenorhabditis Elegans [70]. Since, they have been increasingly studied, and in 2002, Dr. Croce’s group provided the first evidence of miRNA’s involvement in the pathogenesis of human cancer. There is growing evidence on the role of miRNAs in the development, progression and metastasis of various cancers [71] and in a particular way in NSCLC [72,73,74]. Indeed, in 2008, the discovery of circulating miRNAs in plasma and serum meant that these began to be studied as biomarkers for the diagnosis and treatment of NSCLC [75,76]. In NSCLC, miRNAs act as tumor suppressors or oncogenes to regulate progression and metastasis by modulating their target genes. There are several families of miRNAs that act as tumor suppressor genes such as the Let-7 family, which effectively induces cell cycle arrest and cell death in murine lung cancer cells expressing KRAS (G12D) [77]; miR-34 is a direct proapoptotic transcriptional target of p53. Down-regulated miR-34a expression is often observed in NSCLC, contributing to tumorigenesis by attenuating p53-dependent apoptosis [78]; miR-486 is considered an ideal biomarker in cancer diagnosis [79] targeting components related to insulin growth factor (IGF) signaling and functioning as a tumor suppressor in NSCLC [72]. miR-218, which targets the EMT, Slug and ZEB2 regulators [80], and miR-200 inhibit EMT by targeting ZEB1 and ZEB2. Given the controversial function of miRNAs, there are data in the literature that support their role also as oncogenes in NSCLC including miR-196b, miR-221/222, miR-17/92, miR-21 and miR-224 [81]. 

MiRNAs have been shown to be involved in genetic alterations, epigenetic changes and transcriptional control with a key role in NSCLC. Some miRNAs are inherently involved in NSCLC metastasis processing, so a deep understanding of the miRNA signaling network will help identify therapeutic targets. In addition, miRNAs contribute to the drug resistance of NSCLC. Zhao et al. [82] demonstrated how changes in the nature and amount of miRNA in exosomes are associated with the resistance of NSCLC cells to chemotherapy drugs. Other studies argue that miRNAs are differentially expressed in the exosomes of cisplatin CDDP-resistant NSCLC cells [83]. Furthermore, the abnormal expression of miRNAs has been considered one of the causes of resistance to tyrosine kinase (TKI) inhibitors [84]; for this reason, the detection of miRNAs in biological fluids, such as plasma or serum, could serve as crucial circulating biomarkers for the study of drug resistance through the use of non-invasive methods such as liquid biopsy. A systematic meta-analysis including 6919 lung cancer patients and 7064 controls confirmed the diagnostic significance of circulating miRNAs in lung cancer with a specificity of 82% and sensitivity of 81%, in the detection of lung cancer in the early stages (I-II). Furthermore, circulating serum miRNAs appear to be more suitable than those extracted from plasma with superior diagnostic capability in the Caucasian population [85]. However, the application of circulating miRNAs in clinical routine is still hampered by the heterogeneity of the studies, the limited sample size and the lack of perspective and validation studies. The data are not fully reproducible due to hemolysis, the difficulty of RNA isolation, the lack of data normalization and the use of different technological and statistical data analysis platforms. At the same time, miRNAs remain very important biomarkers for the early diagnosis of NSCLC and its clinical management, but further validation studies are needed before this resource can be used in the clinical routine. 

### 3.4. Exosomes in NSCLC

The exosomes are nano-sized vesicles with a diameter of about 30–200 nm. They are released during the fusion processes of plasma membranes through active processes of exocytosis [86,87]. A cancer cell can actively release more than 20,000 exosomal vesicles in 48 h [88], which contain RNA, DNA and proteins that represent the information biological load of cancer cells [89,90,91] contrary to the release of ctDNA and miRNA derived from “dying” cells in necrosis or apoptosis [92]. These differences in secretion point out that the information on exosomes comes from living cancer cells, suggesting a possibility of diagnosing lesions earlier. There is evidence showing that exosomes are closely related to lung carcinogenesis by playing a key role in the growth and progression of lung malignancies through tumor angiogenesis and EMT [93]. Several studies have demonstrated how the use of exosomes and exosomal RNA, combined with ctDNA and proteins, is giving rise to a multi-analyte approach that is very advantageous in terms of sensitivity to the detection of mutations [94,95,96,97,98]. This is demonstrated in Krug AK and colleagues’ studies [96,97], in which they reported how it is possible to obtain up to 10 times more copies of the mutation by combining the data coming from both the exosomal RNA and the ctDNA compared to the single use of ctDNA, allowing detection of the disease in the early stages. Moreover, the combined use of biomarkers is useful in the treatment outcome of NSCLC [97]. The liquid biopsy using exosomes as substrates is a field in exploration, and even if exosomes represent an interesting alternative and/or complement to other forms of liquid biopsy for a better overall diagnostic performance, there are still several challenges to be faced to overcome the low sensitivity of technologies in detecting the high heterogeneity of different subsets of exosomes. Table 1 shows the current advantages and challenges of using different substrates in liquid biopsy techniques. Each of the substrates has advantages and challenges to overcome in order to be fully utilized in clinical practice. What is certain is that each of them can be suitable as a non-invasive oncological investigation for precision medicine, but none of these can give an exhaustive answer. For this reason, the use of these in combination with each other and not as mutually exclusive must be taken into consideration.

### 3.5. Liquid Biopsy in Pleural Effusion

For the purpose of promising precision medicine, confining the study of cfDNA or CTCs and extracellular vesicles to the bloodstream can be limiting. Indeed, malignant pleural effusion (MPE) is rich in tumor cells for patients with advanced lung cancer. The extracellular vesicles and the cfDNA are the two main targets currently explored using MPE. 

The MPE is an accumulation of extra fluid in the pleural space between the lungs and chest wall and its use may have great clinical significance [98]. The MPE is common in patients with metastatic lung cancer with a frequency ranging from 8% to 15% and with a higher incidence for lung adenocarcinoma due to its peripheral location in the lung, being able to invade the pleural space more easily [99]. As a blood sampling, the MPE is also easy to collect and it is more informative for mutation rates than tissue biopsy samples [100]. Several studies have demonstrated that the yield of cfDNA in MPE is higher than that of cfDNA from plasma samples with median concentrations of 278.1 ng/mL and 20.4 ng/mL for MPE and plasma, respectively [101]. However, it seems that MPE has been observed in the advanced stages of cancer, which makes this approach useless for the early detection of lung cancer [101]. cfDNA, miRNA and exosomes offer a good alternative for genomic profiling of pleural effusion when tumor tissue samples are not available and when bloodstream concentrations of these substrates are not optimal for molecular analyses. Thus, MPE is a potential resource for liquid biopsy in the investigation of advanced lung cancer. 

The characteristics of each substrate used as different approaches to liquid biopsy are summarized and compared in Table 2 in which the sensitivity, specificity and odds ratio between plasma and pleural effusion can be appreciated. The liquid biopsy technique that shows the greatest sensitivity and specificity of 96,8% and 87%, respectively, is that of exosomes analysis extracted from the pleural effusion.

## 4. Conclusions

In this review, we aimed to establish where we are and in what direction we are going by collecting data from the literature about the different types of liquid biopsy used for the study of NSCLC. Liquid biopsy is useful for lung cancer management with a role in real-time and long-term prognosis, diagnosis, screening and monitoring. We have seen how the single use of cfDNA has a limit in the detection of mutations in the early stages of NSCLC that can be overcome by combining the analysis with other circulating biomarkers. Therefore, if implemented in daily practice, CTC, cfDNA, miRNA and exosome tests would greatly improve cancer treatment. The development of new molecular diagnostic tools allows for more widespread use of already approved targeted therapies and offers the right treatment to each patient in the best possible way. At present, the literature data are quite controversial, and this makes clear the need for standardized protocols for sample collection and data analysis. However, we must be aware that tissue biopsy and liquid biopsy are not competing, and while tissue biopsy is still the diagnostic gold standard, liquid biopsy is another viable mutation detection option.

## Figures and Tables

**Figure 1 medsci-11-00008-f001:**
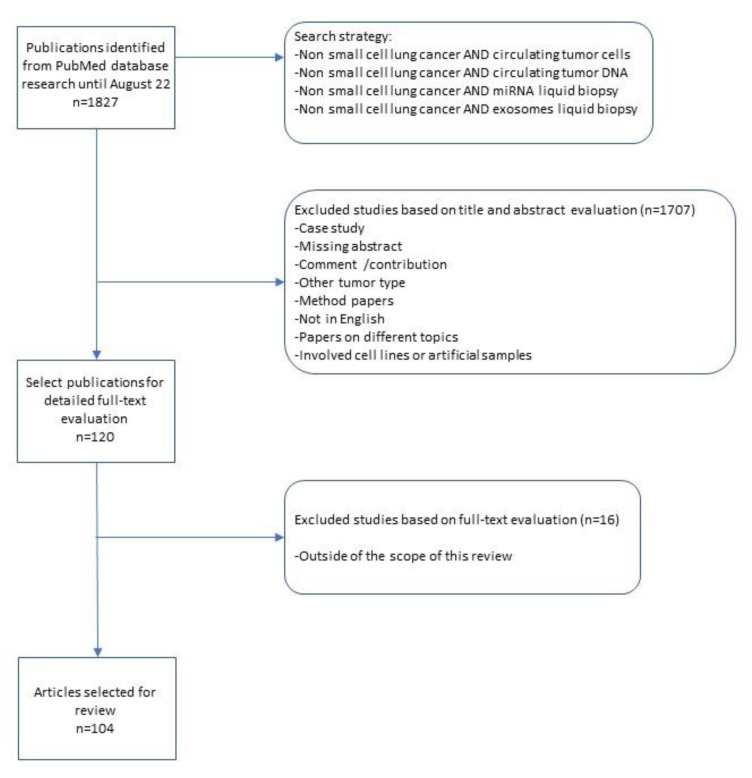
Flowchart of the literature selection.

**Table 1 medsci-11-00008-t001:** Advantages and challenges of single substrates used for liquid biopsy.

Substrate	Advantages	Challenges
**cfDNA**	- Easy to isolate - Stable - Proportion related to tumor burden and overall survival	- Prior knowledge of the target of interest - Limitation of available material - Blood cell death under therapy could spike cfDNA fraction - Source not clear - Large background of ”normal” cfDNA
**CTCs**	- Prognostic in all stages - Matching mutation with tumor subclones - Intact cells could be resistant clones - Functional assays - Culture	- Difficult to isolate - Sampling bias of captured cells - Single-cell/low cell number sequencing challenging
**miRNA**	- Early diagnosis - Predictors of diagnosis, prognosis and response to therapies - High accuracy, sensitivity and specificity - Associated with overall survival	- Not stable - High variability - miRNA detection and quantification pre-analytical steps not validated - Non-specificity for a type of cancer
**Exosomes**	- Contain various biomolecules (proteins, RNAs and lipids) - Stable - Selection of patients with poor outcomes based on size - More sensitive to the identification of relevant mutations than that of cfDNA.	- Difficult to isolate - No or minimal overlap occurs across the studies, strongly limiting their clinical usefulness.

**Table 2 medsci-11-00008-t002:** Table for evaluating the diagnostic value of single substrates compared between plasma and pleural effusion.

Cases	Controls	Substrates	Sample Type	Sensibility	Specificity	Odds Ratio	References
1193	1059	cfDNA	Plasma	81%	85%	23.87	Jiang T et al. [102]
74	-		Pleural effusion	88%	100%	-	Kawahara A et al. [103]
460	239	CTCs	Plasma	75%	92%		Huang H et al. [104]
-	-		Pleural effusion	-	-	-	
1187	879	miRNA	Plasma	85%	84%	31.77	Wang Z et al. [105]
-	-		Pleural effusion	-	-	-	
1338	1075	Exosomes	Plasma	82%	84%	25.14	Song Z et al. [106]
224	-		Pleural effusion	96.8%	87%	-	Kim IA et al. [107]

## Data Availability

The data presented in this study are available in this article.

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
