# Peer review of "Different Liquid Biopsies for the Management of Non-Small Cell Lung Cancer in the Mutational Oncology Era"

_medsci, 2023, doi:10.3390/medsci11010008_

Round 1

Reviewer 1 Report

In this manuscript, the authors tried to explain the importance of liquid biopsies in NSCLC prognosis and diagnosis. This is a significant clinical issue. However, it would be better to go through it in detail, including sensitivity, specificity, odds ratio, PFS, and OS. The authors might think about selecting several dominant articles for each type of specimen and listing their detailed information in a table. Then, the readers would have a clue for comparison. In addition, there is a lack of a section comparing different types of specimens. Adding this section could be better for reading. Besides, this manuscript generally described blood samples. But in practice, pleural effusion, and lavage samples would be also available in routine checks in NSCLC patients. Pleural effusion aspiration even is widely used in releasing symptoms for late-stage NSCLC patients. The authors could consider adding the liquid biopsies of pleural effusions or lavage. Theoretically, these specimens could contain more tumor-oriented particles or DNA. 

Author Response

Independent Review Report, Reviewer 1

In this manuscript, the authors tried to explain the importance of liquid biopsies in NSCLC prognosis and diagnosis. This is a significant clinical issue. However, it would be better to go through it in detail, including sensitivity, specificity, odds ratio, PFS, and OS. The authors might think about selecting several dominant articles for each type of specimen and listing their detailed information in a table.

A: We thank the reviewer for the comment. The table was mentioned and added at line 318 and a section has been included at lines 314-316

Then, the readers would have a clue for comparison. In addition, there is a lack of a section comparing different types of specimens. Adding this section could be better for reading.

A: Done. A table to answer to this request has been added at line 289 and a section has been included at lines 282-288.

Besides, this manuscript generally described blood samples. But in practice, pleural effusion, and lavage samples would be also available in routine checks in NSCLC patients. Pleural effusion aspiration even is widely used in releasing symptoms for late-stage NSCLC patients. The authors could consider adding the liquid biopsies of pleural effusions or lavage. Theoretically, these specimens could contain more tumor-oriented particles or DNA.

A: Done. A paragraph has been added from line 291 to line 316.

Reviewer 2 Report

I would like to congratulate the authors for addressing this very important and timely topic.  This is an era of patient centered and precision guided therapy.  The liquid lung biopsy are getting available and help with the diagnosis specially in non-small cell lung cancer.

The manuscript introduction and details are excellent.  The authors have covered this important topic in concise and easy to understand. Flow and language quality is good

I do have few minor comments.

1.        I would suggest the method section to be moved up from line 259 to line 130, as methodology always comes before the details regarding the details about the manuscript.

2.       I would suggest consider a table describing the advantage/challenge associated with CT-DNA, MiRNA, Exosome

3.       Discussion heading should be changed to conclusion as that appears more like conclusion.

Author Response

Independent Review Report, Reviewer 2

I would like to congratulate the authors for addressing this very important and timely topic.  This is an era of patient centered and precision guided therapy. The liquid lung biopsy are getting available and help with the diagnosis specially in non-small cell lung cancer.

The manuscript introduction and details are excellent. The authors have covered this important topic in concise and easy to understand. Flow and language quality is good.

I do have few minor comments.

  1. I would suggest the method section to be moved up from line 259 to line 130, as methodology always comes before the details regarding the details about the manuscript.

A: Done.

  1. I would suggest consider a table describing the advantage/challenge associated with CT-DNA, MiRNA, Exosome

A: Done.

  1. Discussion heading should be changed to conclusion as that appears more like conclusion.

A: Done.